# FGPrompt: Fine-grained Goal Prompting for Image-goal Navigation

**Xinyu Sun**[1,3]    **Peihao Chen**[1]    **Jugang Fan**[1]   **Thomas H. Li**[4]
**Jian Chen**[1*]   **Mingkui Tan**[1,2,5*]

[1]South China University of Technology, [2]Pazhou Laboratory,
[3]Information Technology R&D Innovation Center of Peking University,
[4]Peking University Shenzhen Graduate School,
[5]Key Laboratory of Big Data and Intelligent Robot, Ministry of Education,
csxinyusu@gmail.com, mingkuitan@scut.edu.cn
Project Page & Videos: https://xinyusun.github.io/fgprompt-pages

## Abstract

Learning to navigate to an image-specified goal is an important but challenging task for autonomous systems. The agent is required to reason the goal location from where a picture is shot. Existing methods try to solve this problem by learning a navigation policy, which captures semantic features of the goal image and observation image independently and lastly fuses them for predicting a sequence of navigation actions. However, these methods suffer from two major limitations. 1) They may miss detailed information in the goal image, and thus fail to reason the goal location. 2) More critically, it is hard to focus on the goal-relevant regions in the observation image, because they attempt to understand observation without goal conditioning. In this paper, we aim to overcome these limitations by designing a Fine-grained Goal Prompting (FGPrompt) method for image-goal navigation. In particular, we leverage fine-grained and high-resolution feature maps in the goal image as prompts to perform conditioned embedding, which preserves detailed information in the goal image and guides the observation encoder to pay attention to goal-relevant regions. Compared with existing methods on the image-goal navigation benchmark, our method brings significant performance improvement on 3 benchmark datasets (*i.e.,* Gibson, MP3D, and HM3D). Especially on Gibson, we surpass the state-of-the-art success rate by 8% with only 1/50 model size.

## 1   Introduction

We focus on the image-goal navigation (ImageNav) task [51] that requires an agent to navigate to an image-specified goal position and face the same orientation as where the photo is taken. In this task, the agent needs to explore the environment and try to find the objects with their surroundings that best match the ones specified in the goal image. Though humans prefer to share information using language, an image is a much clearer and more detailed description to specify a goal location or an intermediate landmark for some household robots [23] or self-driving vehicles.

Despite its wide applications, this task is still very challenging for the embodied agent due to the following two aspects. First, compared to object-goal navigation which assigns goal descriptions with specific semantic categories, it requires the agent to perceive the visual observation as well as the goal image and make a comprehensive understanding of the scene in order to identify goal-relevant objects. Second, objects share similar semantic meanings within one environment, making it challenging to accurately find out the desired object instance.

---

*Corresponding author.

37th Conference on Neural Information Processing Systems (NeurIPS 2023).

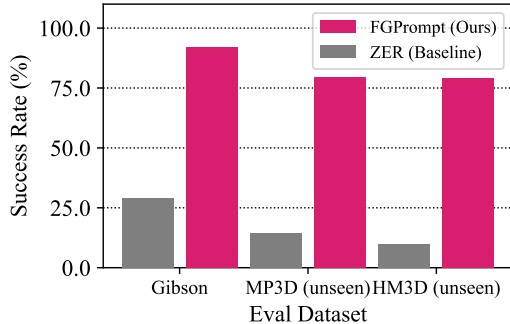 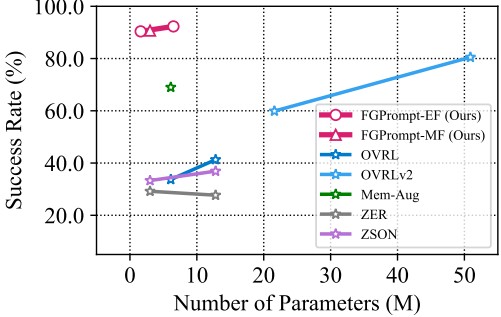

(a) Success rate comparison with *baseline* (ZER [52]) on three different datasets. Our method performs efficiently and robustly in both seen (*i.e.*, Gibson) and unseen (*i.e.*, MP3D and HM3D) environments.

(b) Comparison with SOTA both on success rate and the number of parameters. the FGPrompt-EF, an early fusion variant of our method, achieved 90.4% success rate with only 1/50 model size compared to SOTA.

Figure 1: Main results of our proposed FGPrompt on the image navigation task.

Previous methods [31, 7, 19, 8, 37, 6, 2] seek to solve this task by decomposing the navigation system into several modules in isolation. In general, they tend to adopt efficient exploration skills to build a map incrementally as the understanding of the scene, and further predict a waypoint to navigate to. However, these map-based methods require depth maps or the agent's GPS position to build the occupancy map or topological map. The latest methods [15, 29, 52, 28, 47, 46] instead try to learn a navigation policy in an end-to-end manner using reinforcement learning. These methods set up two different encoders to obtain semantic embeddings from goal and observation images independently. Subsequently, a recurrent model takes these embeddings as input to predict a possible action sequence. However, they suffer from two major limitations: 1) As the details in the goal image are gradually overlooked as it goes through deeper network layers, it is harder to find useful cues for reasoning and finding the goal location. 2) Existing methods leave the goal image apart from the observation when performing encoding, it is hard for the agent to focus on the goal-relevant regions in the observation since there is no goal prompting to guide the agent to understand the observation.

When people try to find a place captured in an image, they must look for the contextual cues presented with objects, shapes, colors, and textures in both the goal images and current visual observation. Spatial reasoning based on this information plays a critical role in understanding the scene, as people always compare and identify similarities, in order to consider the relative position of various elements and gain insights into the current position and the target location. Motivated by this fact, instead of considering only semantic features of goal and observation images, we propose a novel fine-grained goal prompting (**FGPrompt**) architecture to learn observation embeddings conditioned on the fine-grained and high-resolution features of the goal image.

Specifically, we implement the goal prompting scheme as a fusion process between the goal and observation images and design a mid fusion (FGPrompt-MF) mechanism. This mechanism leverages fine-grained and high-resolution feature maps in the intermediate goal network layers as the prompts, which are proven to contain informative object details [20, 50]. Hereafter, conditioned on these feature maps, we utilize FiLM [32] layers to learn a transform function to adjust the observation activations to focus on goal-relevant objects. In addition, we also design an early fusion (FGPrompt-EF) mechanism by concatenating the goal and observation images at the pixel level. We then use a neural network to perform implicit information exchange. Experimental results show that our proposed method significantly outperforms state-of-the-art methods, as shown in Figure 1.

To sum up, our contributions are as follows: 1) We propose a fine-grained image goal prompting (FGPrompt) architecture to explicitly exchange fine-grained information between goal image and observation image, reaching a new SOTA of the ImageNav task and also showing great potential in some other embodied tasks including instance image navigation and visual rearrangement tasks. 2) We empower the agent with fine-grained information exchangeability through a simple channel concatenation technique. This scheme is parameter efficient yet shows an absolute advantage on the ImageNav task, even compared to some complex memory graph-based methods. 3) We dedicately design a mid-fusion scheme through a novel fine-grained FiLM mapping module to perform a more robust information exchange. This scheme shows superior performance in more practical scenarios where the goal image possesses different camera parameters from the observation.

## 2 Related Work

**Modular methods.** Modular methods leverage strictly defined modules that are handcrafted [37, 23] or learnable [7, 19, 18, 8, 37, 6, 2, 10] to address the image-goal navigation task step by step. Classical modular methods typically combine the exploration [48] component, simultaneous localization and mapping (SLAM [16, 43]) component, and path planning component to achieve the navigation goal. In order to localize the agent in an unknown environment, some approaches build an explicit metric map of the environment [7, 19], while others propose to obtain an implicit latent map [18] like a topological map [8, 37] or simply adopt object detectors without mapping [35]. Chaplot *et al.* [6] and Avraham *et al.* [2] train supervised deep models to tackle the sub-tasks, which require a lot of annotated data. Although off-the-shelf modules can be used with zero fine-tuning [23], they still heavily rely on pose and depth sensors, which greatly limits their applicability in the real world.

**RL-based navigation.** Another pipeline for ImageNav is to directly learn from interactions with the environment using reinforcement learning (RL). RL-based navigation tends to learn an end-to-end reward-driven policy that maps observation to action [47, 46, 52, 28, 29, 12] and shows great potential in this task. However, these methods still face the challenge of the sparse reward mechanism and poor generalization performance. To address these issues, previous works [15, 52, 28, 26] propose different methods to encourage the agent to explore more efficiently. Yu *et al.* [15] combines RL policy and visual representation learning model in a min-max game way to incentivize the agent to explore its environment. Al-Halah *et al.* [52] proposes a zero-shot transfer learning approach with a novel reward for its semantic search policy. Similarly, Majumdar *et al.* [28] uses a CLIP model pre-trained in self-supervised manner [33, 41, 9] to enhance image embedding. To tackle the long-horizon planning problem, an external memory module has been proposed by [29, 17, 3, 37, 25, 22, 11] that learns a topological graph [17, 3, 37, 25, 22] or attention map [29] online. Self-supervised learning paradigm has also been explored by Yadav *et al.* [47, 46] to endow the navigation model with better representation ability. Different from existing approaches, we proposed a goal-prompted observation understanding method that learns to focus on goal-relevant objects through fine-grained goal prompts.

**Goal-conditioned learning.** Existing RL-based navigation methods can be interpreted as learning a goal-conditioned policy, since they only perform fusion on the latent goal embedding and observation embedding. Only semantic-level information can be exchanged during fusing. Some embodied robot planning methods [4, 40, 21, 49] learn a goal-conditioned observation encoder by injecting the goal embedding into it. Stone *et al.* [40] and Brohan *et al.* [4] only consider the language as the goal description, while Jang *et al.* [21] and Yu *et al.* [49] try to fuse the goal image with the intermediate feature maps of observation encoder using an affine transformation proposed by FiLM [32]. However, they still focus on the latent embedding of goal images and neglect the fine-grained information in high-resolution activation maps. In this paper, we propose to make use of the intermediate activations in the goal encoder as informative guidance to condition the learning of the observation encoder.

## 3 Image Goal Navigation using Fine-Grained Goal Prompting

### 3.1 Task definition

Image-goal navigation (ImageNav) requires an agent to navigate to a goal position that matches where the goal image $v_g$ was shot. Specifically, the agent starts at a random location $p_0$ and only receives a goal image $v_g$ from the environment. At each time step $t$, the agent receives an egocentric RGB image $v_t$ captured by a RGB sensor fixed on its body, and executes an action $a_t$ conditioned on $v_t$ and $v_g$. In RL-based methods, the action $a_t$ is selected based on the learned policy. After performing the action $a_t$, the agent will be assigned a reward $r_t$ that encourages the agent to reach the goal position as soon as possible. A more detailed definition of our setup can be found in Section 4.

Existing RL-based methods tackle the ImageNav problem by learning an observation encoder and a goal encoder separately, and then fusing their output embeddings together. As shown in Figure 2 (a), this fusion module is commonly equipped on most of the baseline methods. However, those embeddings preserve little detailed information, *e.g.*, shape, texture, and spatial relationship, to promote finding and comparing objects relevant to the goal image [50, 20]. To tackle this challenge, we propose to leverage fine-grained information from lower-level goal image features as prompts to promote the agent's ability to focus more on goal-relevant objects.

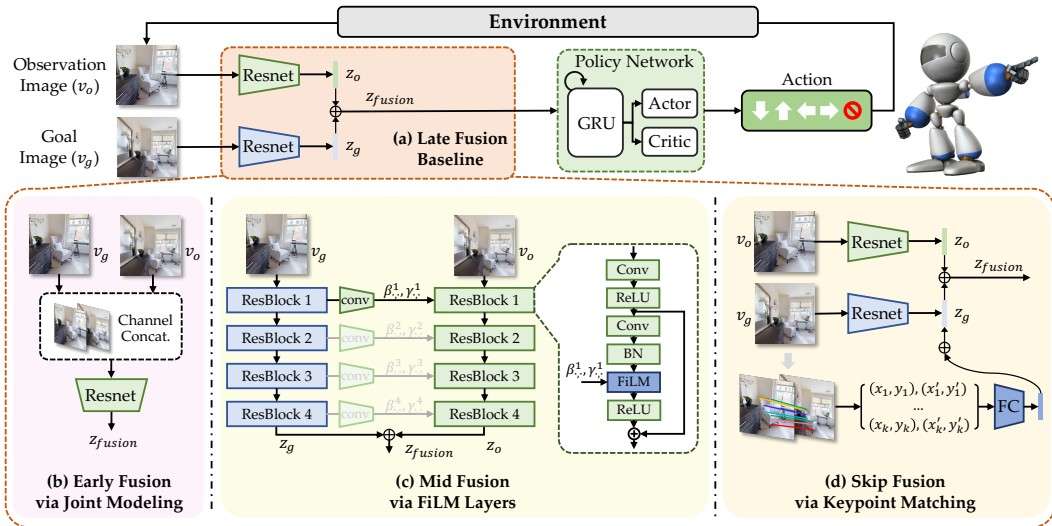

Figure 2: **Illustration of baseline fusion (a) and our goal prompting (b, c, d) for image-goal navigation**. All these methods take observation and goal images as input and output fused features.

## 3.2 Fine-grained Goal Prompting

We design and explore three different fine-grained goal prompting methods that vary from fusion mechanism, namely **Early Fusion**, **Mid Fusion**, and **Skip Fusion**. For the first early fusion mechanism, we investigate injecting fine-grained information from the goal encoder through a simple but effective channel concatenation. After that, we delicately design an explicit information flow through a novel fine-grained FiLM mapping module. Finally, we replace the learnable modules with a heuristic one that injects goal-relative features using feature matching.

**Early Fusion via Joint Modeling.** A naive solution to exchange information in two images is to directly concatenate them together before input to the encoder. In this way, we are able to fuse fine-grained image details in the very early stage and jointly model them using the same encoder. In particular, we concatenate the goal image with the observation image on the RGB channel dimension, resulting in an input tensor shaped $128 \times 128 \times 6$. This concatenated tensor is then fed into a ResNet encoder that takes the 6-channel image as input. In this case, the fusion mechanism can be written as:

$$z_{fusion} = f_o(v_o \oplus v_g) \tag{1}$$

This simple design yields a promising performance on the image navigation benchmark. Detailed ablation on this early fusion operation can be found in Section 4.2.

**Mid Fusion via FiLM Layers.** However, as the early fusion mechanism enables spatial reasoning between two images using an identical convolution kernel, it is difficult to handle the situation when the orientation of the goal camera is noisy. To alleviate this, we further propose an active fusion scheme, utilizing the adaptability of a novel fine-grained FiLM mapping module. Previous literature [21, 49] inputs the goal embedding into the ResNet visual backbone via FiLM [32] layers, which adapt a learnable affine transformation conditioned on the input embedding to the intermediate activation maps in each residual blocks. Through these layers, we can easily connect the intermediate layers in both the goal encoder and the observation encoder to perform mid fusion.

Different from the existing approaches that leverage abstract language embedding as a global condition for all layers, we propose to use the hierarchical representations from the intermediate goal encoder layers. This allows us to make good use of the fine-grained information in high-resolution feature maps. Specifically, we perform FiLM affine transformation on the resnet blocks of the observation encoder, where the affine factors $\beta^i_{\cdot,\cdot}, \gamma^i_{\cdot,\cdot}$ in block $i$ are conditioned on the shaped activation map $z^i_g$ from the correspondent block of the goal encoder. This process can be formulated as:

$$\gamma^i_c = f_c(z_g) \quad \beta^i_c = h_c(z_g) \tag{2}$$

$$\hat{z}^i_o = \gamma^i_c z^i_o + \beta^i_c \tag{3}$$

where $\hat{z}_o^i$ denotes a transformed activation map in block $i$ and $c$ denotes the $c^{th}$ feature of the feature map. The functions $f$ and $h$ learn to map the condition variable into the affine factors. In practice, we implement them as $1 \times 1$ convolutions to maintain the same resolution between the input and target activation map. Section 4.2 further investigates the choices of the mapping function and the number of FiLM layers. The output from the conditioned observation encoder $f_o$ can then be viewed as the fused feature $z_{fusion}$, as shown in Figure 2 (c). The fused feature can be written as:

$$z_{fusion} = f_o(v_o|v_g) \tag{4}$$

Our experiments in Section 4.3 reveal that the mid fusion scheme performs more robustly when the configuration of the goal camera and the observation camera is not perfectly matched.

**Skip Fusion via Keypoint Matching.** In order to evaluate the importance of the addition of the fusion modules, We finally replaced the aforementioned learnable modules with a heuristic one. To achieve this, we follow the idea of Wasserman *et al.* [44] that attach an additional low-level fusion module using handcrafted keypoint matching methods [27, 36], as an improvement of the Late Fusion baseline. We name this mechanism Skip Fusion as it fuses the goal image and observation image in the both early and later stages but skip the others, as shown in Figure 2 (b).

Keypoint matching, which aims to discover representative keypoints in an image and then describe and match them with the most similar ones in another image. As these points are detected based on the low-level statistic [27, 13] of image pixels, we leverage them to play a role as low-level fusion. This scheme is handcrafted as it is not learnable during training. To enable batch inference, we leverage a deep learning-based keypoint detecting [14] and a matching [36] method to obtain matched keypoint between the goal image and the observation image. Hereafter, we select top-k matched points according to their matching score to compose a variable $z_k$ and concatenate them together with $z_g$ and $z_o$ as the fusion result:

$$z_{fusion} = z_g \oplus z_o \oplus \mathrm{FC}(z_k) \tag{5}$$

where $z_k = (x_1, y_1, x_1', y_1', ..., x_k, y_k, x_k', y_k')$ is a flattened vector of $k$ keypoints and FC denotes to a fully connected layer. The default value of vector $z_k$ is set to $-1$ if the number of matched keypoints is less than k. In Section 4.2, we show the superiority of our proposed Early Fusion and Mid Fusion schemes against this heuristic fusion baseline.

### 3.3 Navigation Policy

Based on the fused embedding $z_{fusion}$ of the goal image and observation image, we train a navigation policy $\pi$ using reinforcement learning (RL):

$$s_t = \pi(z_{fusion} \oplus a_{t-1}|h_{t-1}) \tag{6}$$

where $s_t$ is the embedding of the agent's current state. $h_{t-1}$ denotes hidden state of the recurrent layers in policy $\pi$ from previous step. Following previous methods [52, 28], we adopt an actor-critic network to predict state value $c_t$ and action $a_t$ using $s_t$ and train it end-to-end using PPO [39]. We utilize the ZER reward [52] to encourage the agent to not only reach the goal position but also face the goal orientation. More details can be found in Appendix.

## 4 Experiments

**Datasets.** As for image-goal navigation, we use the Habitat simulator [38, 42] and train our agent on the Gibson dataset with 72 training scenes and 14 testing scenes under the standard setting. We use the training episodes provided by [29] and train our agent for 500M steps. We report results under multiple datasets to allow direct comparison to various prior works. On the Gibson dataset, we validate our agent on split A generated by [29], and split B generated by [19]. On the MP3D and HM3D, we use the test episodes collected by [52], as well as the instance image navigation dataset released by [24]. We also extend our method to another embodied task named visual rearrangement, where we use the iTHOR simulator and ai2thor-rearrangement 2023 dataset with 80 training scenes, 20 validation scenes and 20 test scenes. Following [45], we train our agent for 75M steps and finally test the best validation checkpoint on the test set.

| Method | Backbone | Pretrain | Sensor(s) | Memory | Split | SPL | SR |
|---|---|---|---|---|---|---|---|
| NTS [8] | ResNet9 | N/A | RGBD+Pose | ✗ | A | 43.0% | 63.0% |
| Act-Neur-SLAM [6] | ResNet9 | N/A | RGB+Pose | ✗ | A | 23.0% | 35.0% |
| SPTM [37] | ResNet9 | N/A | RGB+Pose | ✗ | A | 27.0% | 51.0% |
| ZER [52] | ResNet9 | N/A | RGB | ✗ | A | 21.6% | 29.2% |
| ZSON [28] | ResNet50 | OSD | RGB | ✗ | A | 28.0% | 36.9% |
| OVRL [47] | ResNet50 | OSD | RGB | ✗ | A | 27.0% | 54.2% |
| OVRL-V2 [46] | ViT-Base | HGSP | RGB | ✗ | A | 58.7% | 82.0% |
| **FGPrompt-MF (Ours)** | ResNet9 | N/A | RGB | ✗ | A | 62.1% | 90.7% |
| **FGPrompt-EF (Ours)** | ResNet9 | N/A | RGB | ✗ | A | 66.5% | 90.4% |
| **FGPrompt-EF (Ours)** | ResNet50 | N/A | RGB | ✗ | A | **68.5%** | **92.3%** |
| Mem-Aug [29] | ResNet18 | N/A | 4 RGB | ✓ | A | 56.0% | 69.0% |
| VGM [25] | ResNet18 | N/A | 4 RGB | ✓ | A | 64.0% | 76.0% |
| OVRL [47] | ResNet50 | OSD | 4 RGB | ✗ | A | 62.5% | 79.8% |
| TSGM [22] | ResNet18 | N/A | 4 RGB | ✓ | A | 67.2% | 81.1% |
| **FGPrompt-EF (Ours)** | ResNet9 | N/A | 4 RGB | ✗ | A | **75.0%** | **94.2%** |
| NRNS [19] | ResNet18 | N/A | RGBD | ✗ | B | 12.4% | 24.0% |
| **FGPrompt-EF (Ours)** | ResNet9 | N/A | RGB | ✗ | B | **70.5%** | **93.0%** |

Table 1: **Comparison with state-of-the-art methods on Gibson**. All methods are trained and evaluated both on the Gibson dataset.

| Methods | Backbone | MP3D | | HM3D | |
|---|---|---|---|---|---|
| | | SPL | SR | SPL | SR |
| Mem-Aug [29] | Resnet18 | 3.9% | 6.9% | 3.5% | 1.9% |
| NRNS [19] | Resnet18 | 5.2% | 9.3% | 4.3% | 6.6% |
| ZER [52] | Resnet9 | 10.8% | 14.6% | 6.3% | 9.6% |
| **FGPrompt-MF (Ours)** | Resnet9 | **50.4%** | **77.6%** | **49.6%** | **76.1%** |

Table 2: **Cross-domain evaluation on MP3D and HM3D**. The agent is trained in Gibson environments and directly transferred to new environments for evaluation.

**Agent configuration.** We follow the recipe of previous trails [52, 28, 47] to initialize an agent equipped with only RGB cameras of $128 \times 128$ resolution and $90°$ FOV. When compared with methods that use a panoramic input, we initialize four RGB sensors to the front, left, right, and back directions of the agent, following [29, 47]. The agent's action space is comprised of four discrete actions, including MOVE_FORWARD, TURN_LEFT, TURN_RIGHT, STOP. The minimum units of rotation and forward movement are $30°$ and 0.25m respectively.

**Evaluation metrics.** We report the success rate (SR) and Success weighted by Path Length (SPL) [1], which takes into account path efficiency of the navigation process. An episode is considered successful if the agent stops within 1.0m Euclidean distance from the goal location and the maximum number of steps in an episode is set to 500 as the default setting.

### 4.1 Comparison with State-of-the-art Methods

**Evaluation on Gibson.** In Table 1, we report the ImageNav results on Gibson averaged over three random seeds (the variances of all random seed results are less than 1e-4.). We compare our methods with state-of-the-art methods in two different settings, one takes only one RGB sensor as input following [52, 28, 47] and another one takes 4 RGB sensors to assemble a panoramic view following [29, 47]. For the SLAM-based methods in the first three rows, we report the number reproduced by Mezghani *et al.* [29]. We found that our proposed FGPrompt-MF and FGPrompt-EF methods take an absolute advantage compared with all previous methods. Even compared to OVRL-V2 [46], a method that utilizes a much larger visual backbone (ViT-B) pre-trained on an in-domain image dataset, we still achieved large performance gains on both SR (92.3% vs. 82.0%) and SPL (68.5% vs. 58.7%) in the absence of additional pose sensor input. This finding reveals the effectiveness and efficiency of our proposed method.

| Setting | SPL | SR |
|---|---|---|
| Later Fusion (baseline) | 11.2% | 13.0% |
| Skip Fusion via keypoint matching (FGPrompt-SF) | 24.7% | 41.6% |
| Mid Fusion via FiLM layers (FGPrompt-MF) | 50.4% | 77.3% |
| Early Fusion via joint modeling (FGPrompt-EF) | **54.7%** | **78.9%** |

Table 3: **Comparison of different goal prompting methods on Gibson ImageNav task**. Fusing the fine-grained goal prompts with the observation instead of directly concatenating their semantic embeddings yield significant improvement.

| Mapping Method | SPL | SR |
|---|---|---|
| N/A | 11.2% | 13.0% |
| Semantic Mapping | 24.0% | 32.0% |
| FG/HR Mapping | **50.4%** | **77.3%** |

| Depth | SPL | SR |
|---|---|---|
| 1 | **50.4%** | 77.3% |
| 2 | 49.3% | **77.6%** |
| 4 | 50.2% | 71.4% |

Table 4: **How to map activation into affine factors?** Using Fine-grained High-resolution (FG/HR) mapping performs significantly better.

Table 5: **How deep should the Mid Fusion perform?** Performing Mid Fusion on the early layers works better than on all layers.

We extend our FGPrompt-EF to the panoramic view setting (4 RGB) for direct comparison with some memory-based methods [29, 25, 22] and pre-trained method [47]. We found that our FGPrompt-EF outperforms these memory-based methods by at least 13.1% in success rate and 7.8% in SPL, even without additional external memory module and pre-training phase. Besides, we also provide a comparison result on the non-mainstream testing episodes (split B) following [19]. Compared with the self-supervised method NRNS [19] that pretrained on passive videos, our FGPrompt-EF brings 58.1% improvement in success rate and 69.0% in SPL, which shows a great advantage by learning to understand the scene based on goal prompting through interacting with the environment.

**Cross-domain evaluation on out-of-domain datasets.** In Table 2, we report the cross-domain evaluation results on the unseen scenes in the Matterport3D (MP3D) [5] and HM3D [34] to verify the generalization ability. Following [52], we directly transfer our model trained on Gibson to these two new datasets, without any tuning. Since there exists a very large domain gap between these datasets (*e.g.* more complex and larger scenes in MP3D and diverse scene types in HM3D), this setting is extremely challenging. We leverage the testing episodes released by ZER [52]. Compared with the baseline method ZER, our fine-grained goal prompting method brings $7\times$ improvements in the success rate, which shows the generalization ability of our method.

## 4.2 Ablation Study

In this section, we first compare the effectiveness of different variants of our method on the ImageNav task. Then we present the detailed ablation of each method to empirically discover their best implementation. For convenience and fairness, all variants in the ablation study are trained for 50M steps on the Gibson dataset.

**Comparing different goal prompting methods.** We first compare the proposed goal prompting variants on the image-goal navigation task. As shown in Table 3, the Skip Fusion (FGPrompt-SF) variant, integrated fine-grained information by simply adding a keypoint matching-based fusion module to the baseline, performs significantly better (from 14.0% to 41.4%). This reveals that fine-grained goal prompting is important. However, this heuristic method does not work when there is no matched area in the observation. The other two variants further tackle this problem by learning a joint-modeling framework. In detail, the Mid Fusion (FGPrompt-MF) mechanism leverages the intermediate activation maps with varied resolutions to perform goal prompting. As a result, this variant further increases the navigation success rate by 27.2%. Besides, as a simplified version of our proposed Mid Fusion mechanism, the Early Fusion mechanism enables an implicit fusion process through jointly modeling the goal and observation images. In Table 3, this simple but ingenious design brings a further improvement (4.3% in SPL) compared to the Mid Fusion mechanism which is well-designed and ablated. We attribute this to its adaptive and learnable fusion fashion.

| Setting | SPL | SR |
|---|---|---|
| 3D stack | 17.3% | 20.5% |
| Edge concat | 37.2% | 54.8% |
| Channel concat | **54.7%** | **78.9%** |

Table 6: **How to perform early fusion?** A naive concatenation at the channel dimension works the best.

| Setting | SPL | SR |
|---|---|---|
| Separate modeling | 11.2% | 13.0% |
| Tied modeling | 12.3% | 14.6% |
| Joint modeling | **54.7%** | **78.9%** |

Table 7: **Does joint modeling works?** Yes, it greatly boosts navigation performance compared to the baseline and another similar approach.

| Method | SPL | SR |
|---|---|---|
| Baseline (no fusion) | 10.5% | 12.1% |
| FGPrompt-EF (Ours) | **38.5%** | **64.6%** |
| FGPrompt-MF (Ours) | **42.5%** | **70.2%** |

Table 8: **Evaluation on the augmented image navigation episodes.** Our mid-fusion mechanism is more robust under the dynamic camera parameter setting.

| Method | SPL | SR |
|---|---|---|
| Baseline (no fusion) | 0.2% | 0.6% |
| FGPrompt-EF (Ours) | **0.8%** | **3.4%** |
| FGPrompt-MF (Ours) | **2.8%** | **9.9%** |

Table 9: **Evaluation on the instance image navigation dataset.** Our mid-fusion mechanism performs better than the baseline image navigation method and the early-fusion variant.

**Ablation on the Mid Fusion mechanism.** We further investigate the detailed setting of our proposed Mid Fusion mechanism. We conduct ablation studies on the design of FiLM layers in Table 4. We design two different mapping methods to map the activation map into the affine factors in Equation 2, namely Semantic Mapping and Fine-grained High-resolution Mapping. Specifically, for the former, we average pool the activation map in each layer to remove the fine-grained information. For the second method, we keep the spatial resolution of the original activation maps, hence preserving the fine-grained information. We initialize two convolution layers with $1 \times 1$ stride to learn a mapping function. Not surprisingly, only taking the coarse-grained input from the goal encoder as a condition leg a lot behind, as it lose lots of details that might serve as possible cues during the pooling.

Another important question is how deep the network layers should be considered to perform fusion. Since the perception field grows as the feature map resolution reduces in deeper layers, the information about objects and scenes in these layers could be more and more coarse-grained. We design an ablation study that integrates a different number of network layers to perform fusion. As shown in Table 5, we found that fusing the first two network layers (each layer indicates an entire Resnet block) performs well, indicating that fine-grained information in the early layers is important for goal prompting. When the fusion depth increases to 4 layers, the navigation performance slightly degrades, as considering more prompting layers increases the learning difficulty.

**Ablation on the Early Fusion mechanism.** Finally, we conduct an ablation study to find out how to perform early fusion on the goal image and observation image. There exists a naive approach to merging them at the pixel level. In particular, we try to concatenate these two images on different dimensions, as shown in Table 6, where concatenation on the channel dimension performs better than on edges (*e.g.* H and W). We conjecture that aligning and modeling the goal and observation images enables better spatial reasoning. We also investigate stacking images at an additional axis and performing 3D convolution to embed them together. Interestingly, results show that this variant failed to learn an effective fusion process, although it aligns both images in the spatial dimension.

We then compare the early fusion scheme with a similar approach that shares the same parameters between the goal encoder and observation encoder following [29], namely Tied Modeling. In Table 7 we directly compare them with a baseline that learns a goal encoder and an observation encoder separately. We observe that the Tied Modeling variant performs worse similar to the Separate Modeling baseline. Though using shared parameters to encode both goal and observation images, this architecture does not enable goal-prompted learning to focus on the goal-relevant regions and thus failed to effectively reason the goal position.

### 4.3 Analysis and Qualitative Visualizations

**Robustness under dynamic camera parameter setting.** We first test the agent trained on imagenav dataset under the dynamic camera parameter setting. We randomly augment the camera height, pitch angle, and HFOV of the goal image in Gibson ImageNav eval episodes. Specifically, we follow the

| Method | Success↑ | FixedStrict↑ | E↓ |
|---|---|---|---|
| ResNet18+IL Basline [45] | 1.89 | 4.92 | 1.32 |
| Ours | **10.2** (+439%) | **24.9** (+406%) | **0.81** |

Table 10: **Results on AI2THOR 1-Phase Rearrangement Challenge.** We apply our proposed FGPrompt method on an imitation learning baseline with a ResNet18 backbone. Surprisingly, we found that our inserted module significantly improved the agent's performance on the 1-Phase track of the visual rearrangement task.

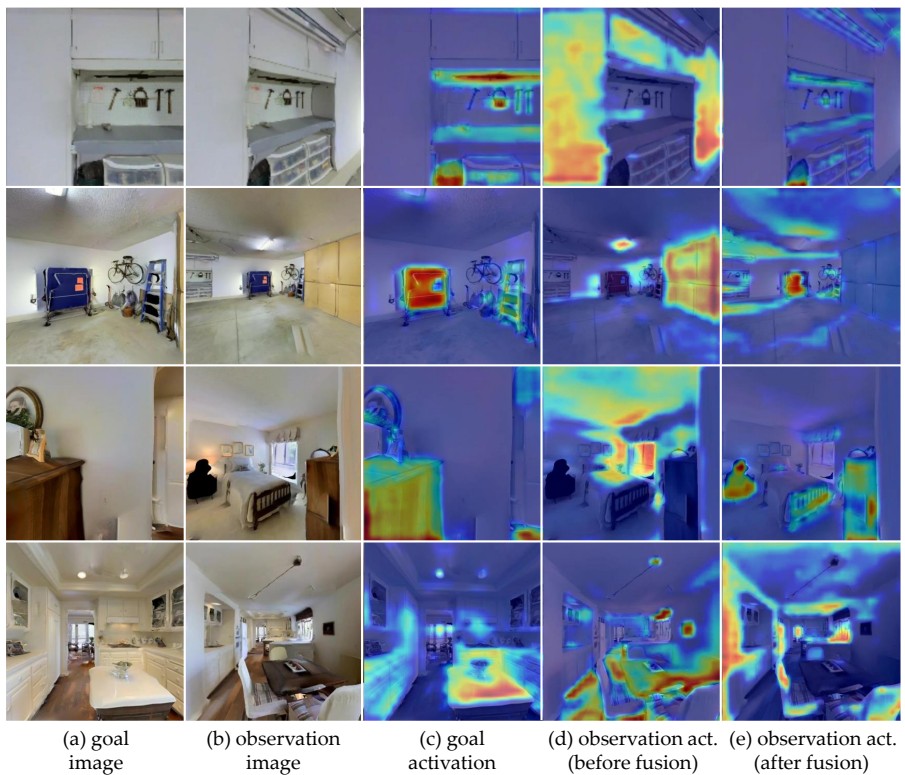

| (a) goal image | (b) observation image | (c) goal activation | (d) observation act. (before fusion) | (e) observation act. (after fusion) |

Figure 3: **EigenCAM visualization of the activation map in the fusion layer of FGPrompt-MF.** Images in different rows illustrate results in different testing episodes in Gibson. The Mid Fusion mechanism learns to focus on the objects that are relevant to the goal image.

distribution of these parameters in the instance imagenav paper [24], sampling goal camera height $h \sim \mathcal{U}(0.8m, 1.5m)$, pitch delta from $\mathcal{U}(-5°, 5°)$, and HFOV from $\mathcal{U}(60°, 120°)$. In Table 8, we find that the mid-fusion mechanism performs the best in this scenario.

We further conduct experiments on the instance image navigation (iin) dataset collected by [24], The episodes in this dataset cover a wide range of object instances in the environment and are much harder to finish. We train three agents on the HM3D ImageNav dataset and evaluate them on the test split of the iin dataset. In Table 9, the baseline model performs poorly in this task with a very low success rate (less than 1%). The agents with our proposed fusion mechanisms both perform better. We also observed that the mid-fusion variant outperforms early fusion in this scenario, as its delicately designed activation deformation module yields explicit and adaptive guidance from the goal image. All these results reveal the robustness of mid-fusion in harder tasks.

From Table 9, the performance of our methods on the instance imagenav task is relatively low compared to the ImageNav task. We speculate that the extremely different perspective of goal images that haven't been seen during training and a longer episode length undermine the performance of our method. This result hints that our method could make a further improvement when combined with memory-based methods [25, 22] to achieve more efficient large-scope exploration.

From the experimental results above, we observed a trade-off between two different fusion schemes. The early-fusion scheme is somehow an interesting finding in that it performs competitively and has a simpler architecture. However, though performs well on the default setting, it does not generalize well to other scenarios where the goal camera parameters don't match with the agent's one. In contrast, our delicately designed mid-fusion mechanism performs better in this case. These results indicate that a carefully designed mid-fusion scheme with more inductive bias is necessary.

**Transfer to the visual rearrangement task.** To see whether our FGPrompt have wide application scenarios, we extend our method to visual rearrangement, another embodied challenge, which aims to move the objects to a correct position in the environment according to unshuffled images. We conduct experiments on the 1-Phase track of the ai2thor-rearrangement challenge and find our method useful in this task. We start from a ResNet18+IL baseline that separately encodes the unshuffled image and agent's current observation without a fusion mechanism and learn from expert actions. Then we introduce our proposed FGPrompt-EF module into the baseline model by fusing the observation with the unshuffled image in an early stage, resulting in one jointly modeled ResNet encoder. We train and test both methods on 2023 dataset and follow [45] to report the testing metrics of the best checkpoints in Table 10. Our proposed module brings 400% relative improvement compared to the baseline. We believe it helps the agent to locate correspondent or inconsistent objects in the environment.

**How does the fine-grained goal prompting work?** We visualize the activation maps using Eigen-CAM [30] before and after the fusion layers of our mid fusion goal prompting method (FGPrompt-MF) to find out how it works in the image navigation task. Illustrations are presented in Figure 3. Prompted with the fine-grained and high-resolution activation map from the goal image, the agent is able to find out the relevant objects in the current observation and pay more attention to them, as shown in the activation visualization in the last column. Interestingly, we found that even though the agent is far away from the goal position, the mid fusion mechanism still guided the observation encoder to focus on relevant objects or explore some candidate regions that may contain the target objects (see the *kitchen bar* in the last row). We also provide visualization and analysis of the other two goal prompting methods in Appendix.

**Performance versus model size.** To discuss the feasibility of application on real-world robot systems with resource-limited devices (*e.g.*, household robots), we investigate and compare the model size of our models with previous ones. We report the agent's number of parameters, as well as the ImageNav success rate on Gibson, and visualize them on the same coordinate system. As shown in Figure 1b, our FGPrompt-EF model outperforms existing methods by a large margin with a much smaller model size, indicating its promising ability on applying to real-world robot systems.

## 5  Discussion

**Limitation and future work** Although our proposed FGPrompt achieved great improvements on different ImageNav datasets, we still need a comprehensive study to find out if this method is applicable to real-world robots. In the future, we will investigate how to deploy this visual navigation methodology to a real-world robot system, to perform sim-to-real transformation.

**Conclusion** In this paper, we propose a novel fine-grained goal prompting (FGPrompt) method for visual navigation. In particular, we design a Mid Fusion architecture via FiLM Layers conditioning (FGPrompt-MF), which leverages the high-resolution activation maps from the goal encoder to perform an affine transformation on the observation encoder. Furthermore, we rethink it and condense it into an Early Fusion mechanism via joint modeling (FGPrompt-EF), with implicit learning of the fusion process. Experimental results on the Image-goal Navigation task show our method has excellent performance, concise architecture design, and strong generalization ability to unseen environments.

**Acknowledgments**

This work was partially supported by the National Natural Science Foundation of China (Grant No. 62072190 & 62376099 & 62072186), the Guangdong Basic and Applied Basic Research Foundation (Grant No. 2019B1515130001), the Program for Guangdong Introducing Innovative and Enterpreneurial Teams 2017ZT07X183, the CCF-Tencent Open Fund RAGR20220108.

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

# Appendix for
# "FGPrompt: Fine-grained Goal Prompting
# for Image-goal Navigation"

In the appendix, we provide more implementation details and experimental results of our FGPrompt. We organize the appendix as follows.

- In Sec. A, we provide more architecture details on three different types of goal-prompting methods.
- In Sec. B, we provide more experimental details, *i.e.*, training and evaluation settings.
- In Sec. C, we provide more ablation results on the skip fusion mechanism.
- In Sec. D, we provide more ablation results on the mid fusion mechanism.
- In Sec. E, we provide a direct comparison with the memory-based methods.
- In Sec. F, we provide additional visualization results.

## A  Architecture details

**Skip fusion mechanism.**    As discussed in the previous sections, we design a skip fusion mechanism that utilizes keypoint detection and matching method to provide the agent with low-level prompting. In order to increase the training speed, we abandon the handcrafted local feature descriptor and detector, instead of a deep learning-based keypoint matching pipeline [2, 10], as shown in Figure 1. A convolution-based keypoint detector is adopted to extract keypoints from both the goal image and observation image. After that, a 9-layer graph neural network with attention modules is applied to find the paired points that share similar features.

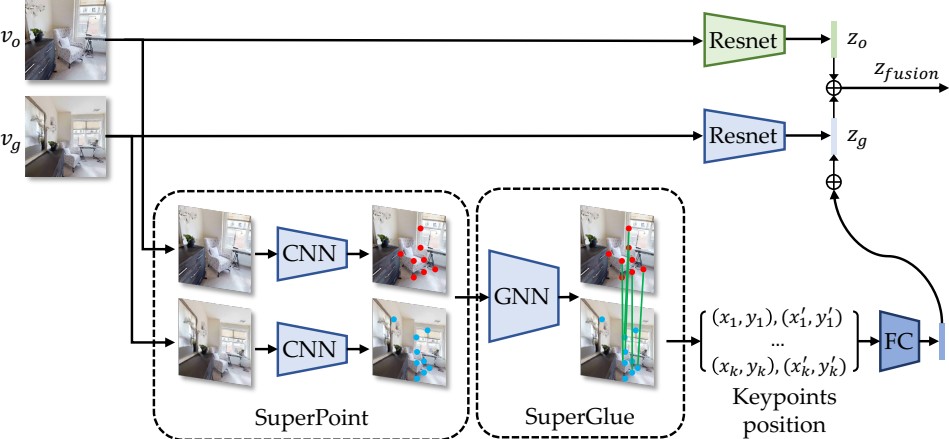

Figure 1: Architecture details of the skip fusion mechanism.

**Mid fusion mechanism.**    We illustrate the detailed architecture of the proposed mid fusion mechanism in Figure 2. For a Resnet9 backbone, we map the intermediate activation maps of each resnet block (ResBlock) into affine factor $\beta$ and $\gamma$ using a $1 \times 1$ convolution and a fully connected layer. Then the $\beta$ and $\gamma$ are injected into the correspondent ResBlock in the observation encoder, guiding the model to focus on goal-relevant regions in the observation.

**Early fusion mechanism.**    The early fusion mechanism is initialized as a single Resnet9 [12] encoder. The stem convolution layer takes a $128 \times 128 \times 6$ image as input. The following layers have no difference from a standard Resnet encoder. We conduct experiments using both Resnet9 and Resnet50 encoders.

**Navigation policy.**    We initialize the navigation policy network as a 2-layer GRU with an embedding size of 128.

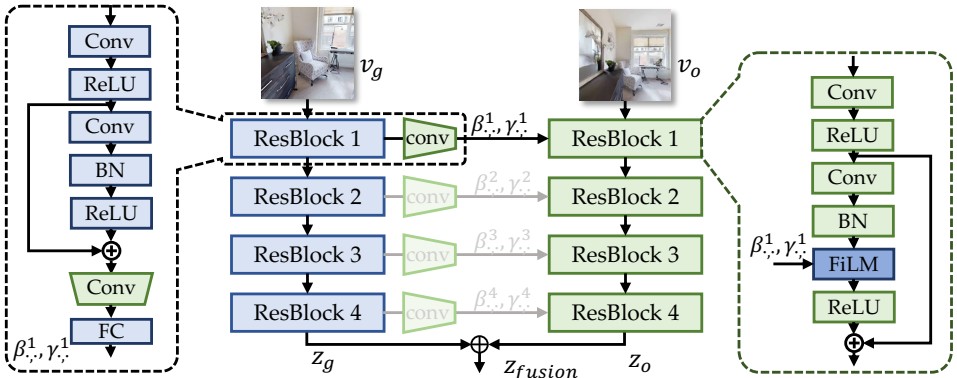

Figure 2: Architecture details of the mid fusion mechanism.

## B   Experimental details

**Dataset details.**   We train our agent on the Gibson dataset and validate the agent on the Gibson, MP3D, HM3D datasets respectively. For the training dataset, there are 72 scenes in total, each scene has 9k episodes, resulting in 648k episodes. The 9k episodes in each scene are evenly divided into three levels according to the distance from the start location to the goal location: easy (1.5 - 3m), medium (3 - 5m), and hard (5 - 10m). For evalution on Gibson, we use two split, in which split A [8] has 14 scenes with 1.4k episodes per level and split B [3] has 14 scenes with 1k episodes per level. For evalution on MP3D and HM3d, we use the same test splits as [12], which has 100 scenes with 1k episodes per level and 18 scenes with 1k episodes per level respectively.

**Training & validation details.**   We use the Habitat simulator to train our model on the Gibson dataset using 20 environments running in parallel with $8\times3090$ GPUs. We set the total training time steps to 500M. For one episode, we set the maximum time steps to 500 when performing validation. Other detailed hyperparameters of DD-PPO training follow the recipe of ZER [12].

**Reward**   We use the reward formulation proposed by [12] that consists of three parts, including dense shaping reward $r_{ds}$, dense slack reward $\gamma$, and sparse success reward $r_{ss}$. The dense shaping reward is defined as:

$$r_{ds} = r_d(d_t, d_{t-1}) + [d_t \leq d_s]r_\alpha(\alpha_t, \alpha_{t-1}), \tag{1}$$

$$\text{where } [A] = \begin{cases} 1, & \text{if A is True} \\ 0, & \text{if A is False} \end{cases} \tag{2}$$

where $r_d$ is the reduced distance to the goal from the current position relative to the previous one, and $r_\alpha$ is the reduced angle in radians to the goal view from the current view relative to the previous one. This reward function not only encourages the agent to approach the goal as much as possible, but also encourages the agent to rotate to a view as similar as possible to the goal view when the agent is close enough to the goal. At each time step t, the agent receives a reward $r_t$ composed of shape reward and slack reward:

$$r_t = r_{ds} - \gamma \tag{3}$$

where $\gamma = 0.01$ is the slack penalty that encourages planning a shorter path to the goal. Once predicted a STOP action, the agent will receive a sparse success reward $r_{ss}$ which is determined by its distance and angle to the goal:

$$r_{ss} = 5 \times ([d_t \leq d_s] + [d_t \leq d_s \text{ and } \alpha_t \leq \alpha_s]). \tag{4}$$

Following [12], we set success distance $d_s = 1$m and $\alpha_s = 25°$. As proven by [12, 7, 11], this reward enables the agent to learn to associate between observation $v_o$ and goal image $v_g$. draw the association between its observation ot and the goal IG. Specifically, The agent will get a sparse reward $r_{ss} = 5$ if it is within $d_s$=1m from the goal, and 10 points if it is also within $\alpha_s$=25° from the goal view. Otherwise, it will get a zero reward.

**4 RGB setting.** To compare with some methods that take panoramic images as input, we equip our agent with 4 pairwise orthogonal cameras to obtain a panoramic view. To reduce the computation cost, we only take the front image of these cameras as the goal image. During training and inference, each RGB image is combined with the goal image and input to our FGPrompt-EF model, and we concatenate all outputs as the visual-motor feature.

## C   More ablation study on skip fusion mechanism

As discussed in previous sections, we introduce an image feature matching module to provide the agent with fine-grained low-level goal prompts. A straightforward approach is applying the handcrafted local descriptors [6, 1] to detect and match paired image regions. It first detects the representative keypoints in the image and then matches the paired keypoints. The matched keypoints represent similar regions in two images that are scale-invariant, for example, the corner of a table or a part of a unique texture on a closet. However, computing these handcrafted features is time-consuming, as it requires computing Gaussian differences on different pyramid scales. In practice, this operation does not support high concurrency when training in the simulator and results in low FPS. To tackle this issue, we utilize a deep learning-based keypoint detector and matcher, called SuperPoint [2] and SuperGlue [10], in order to achieve batch inference on GPU devices. We provide the speed comparison in Table 1. To comprehensively explore how can we leverage this low-level information to perform goal prompting, we provide detailed ablation on this module. In Table 2, we compare different representation methods of the matched keypoints, where position denotes combining the normed pixel coordinate of each paired keypoint and descriptors means averaging the 256-dimension feature of each paired keypoint. We found that simply providing the agent with the location of matched points works the best.

| Matching method | Device | FPS |
|---|---|---|
| SIFT | CPU | 20 |
| SuperPoint + SuperGlue | GPU | 400 |

| Method | SPL | SR |
|---|---|---|
| Position | **37.1%** | **52.5%** |
| Descriptors | 22.9% | 38.1% |
| Descriptors + Position | 24.2% | 43.2% |

Table 1: **Comparing the forward speed of different image matching methods.** We report the frame per second (FPS) metric during training in the simulator.

Table 2: **Comparing different representations of the matched keypoints.** Directly combining the position of paired keypoints performs the best.

## D   More ablation study on mid fusion mechanism

**Fusion layer.** We have verified the effectiveness of fusing low-level information using low-level handcrafted descriptors in previous studies. As discussed in previous literature, the intermediate features in the earlier layer of deep convolution networks contain low-level information (*e.g.*, shape, texture, color, *etc.*). In the above ablation studies, we found that fusing these intermediate features using FiLM layers into observation encoder layers works. We further provide a detailed ablation study on the choice of the fusion layer. From Table 3, fusing later layers with coarse-grained contents performs worse than the first layer. These results show the importance of our proposed fine-grained goal prompting method.

**Comparing more mapping schemes.** In the previous sections, we have shown the priority of our proposed fine-grained goal prompting that mapping the intermediate high-resolution activation maps into the affine factors. To further verify the necessity of fine-grained and high-resolution mapping in the mid fusion mechanism, we provide detailed ablation on the semantic mapping methods, where we shift the source of the semantic goal prompt from the average pooled feature of each activation map to a high-dimension feature in the last layer. The poor performance of these two variants in Table 4 further indicates the importance of fine-grained and high-resolution mapping.

**Comparing FiLM with self-attention.** We conduct an experiment that replaces the FiLM module with a self-attention module. Specifically, we project the flattened feature map from the first layer of the goal encoder into the query and the correspondent sequence from the observation encoder

| Layer | SPL | SR |
|---|---|---|
| 1 | **50.4%** | **77.3%** |
| 2 | 44.4% | 69.2% |
| 3 | 45.9% | 67.3% |
| 4 | 37.1% | 52.5% |

Table 3: **Choice of fusion layer.** Early layers contain informative clues for prompting the observation encoder.

| Mapping Method | SPL | SR |
|---|---|---|
| FG/HR | **50.4%** | **77.3%** |
| Semantic (each layer) | 24.0% | 32.0% |
| Semantic (last layer) | 24.4% | 32.3% |

Table 4: **How to perform semantic mapping?** Neither mapping the global mean of each activation layer or semantic-level feature works.

| Methods | SPL | SR |
|---|---|---|
| Self-attention | 12.2% | 13.9% |
| Ours | **50.4%** | **77.3%** |

Table 5: **How to perform mid-fusion?** Self-attention performs significantly worse than FiLM layers.

| Setting | SPL | SR |
|---|---|---|
| Ours w/o background | 45.2% | 64.4% |
| Ours w/ background | **50.4%** | **77.3%** |

Table 6: **Importance of background context.** Removing background context in goal image leads to a performance decrease.

into key and value. Then, we utilize the self-attention operation to merge the goal and observation features. Experiment results are shown in Table 5.

**Importance of environment context in goal image.** We leverage the semantic annotation from 145 scenes in HM3D v2 dataset and set the pixel of background (e.g., uncountable object such as wall and floor) to zero according to the ground truth segmentation map. Numbers are reported in Table 6. From the experimental results, our method suffered from a slight degradation when the environmental context was removed from the goal image. These results indicate that environmental context in the image background provides useful but limited clues. We believe that objects with their arrangement in each room play a critical role in our FGPrompt.

# E  Comparison with memory-based methods

In Table 7, we directly compare our FGPrompt-EF with two different memory-based image navigation methods on the evaluation split B [3]. The reported results are averaged on both straight and curved path types. Although we do not use an additional memory module to store past agent states and model their relationship using the graph neural network, our method still shows priority over the memory-based methods. In our future work, we will discuss the effectiveness of our goal prompting module by involving it with the memory-based methods.

| Method | Backbone | Pretrain | Sensor(s) | Memory | Split | SPL | SR |
|---|---|---|---|---|---|---|---|
| VGM [5] | ResNet18 | N/A | 4 RGB | ✓ | B | 55.1% | 75.3% |
| TSGM [4] | ResNet18 | N/A | 4 RGB | ✓ | B | 76.9% | 85.4% |
| **FGPrompt-EF (Ours)** | ResNet9 | N/A | 4 RGB | ✗ | B | **78.3%** | **96.4%** |

Table 7: **Comparison with memory-based methods on eval split B.**

# F  Additional visualization results

**Training curve vs. validation curve.** We visualize the training and validation curve in both Success Rate and SPL metrics. As shown in Figure 3, our agent does not overfit the training scenes and has a consistent performance on both the training and validation episodes.

**Visualizing FGPrompt-SF.** In Figure 5, we visualize the matching results of the skip fusion mechanism of our proposed FGPrompt-SF. The first row shows a successfully matched example that numerous keypoints are detected and paired. The second and last rows show failure cases that the keypoint matching module makes incorrect predictions or failed to find corresponding regions, respectively. In the case of the agent's observation completely different from the goal image, this

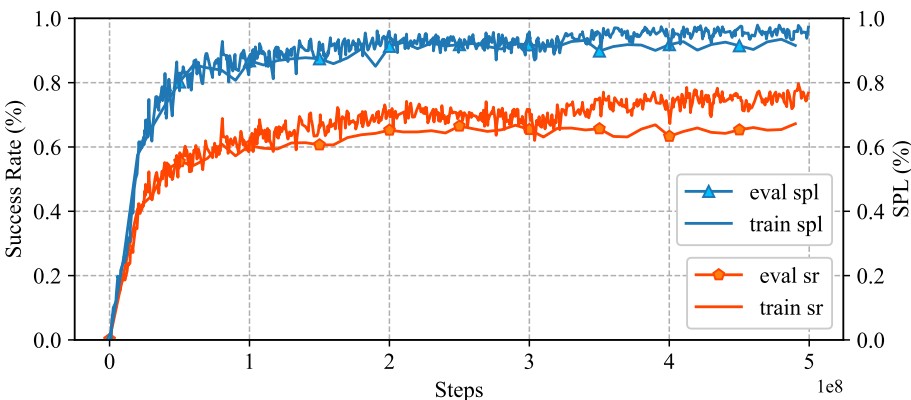

Figure 3: **Training and validation curve in Success Rate and SPL.**

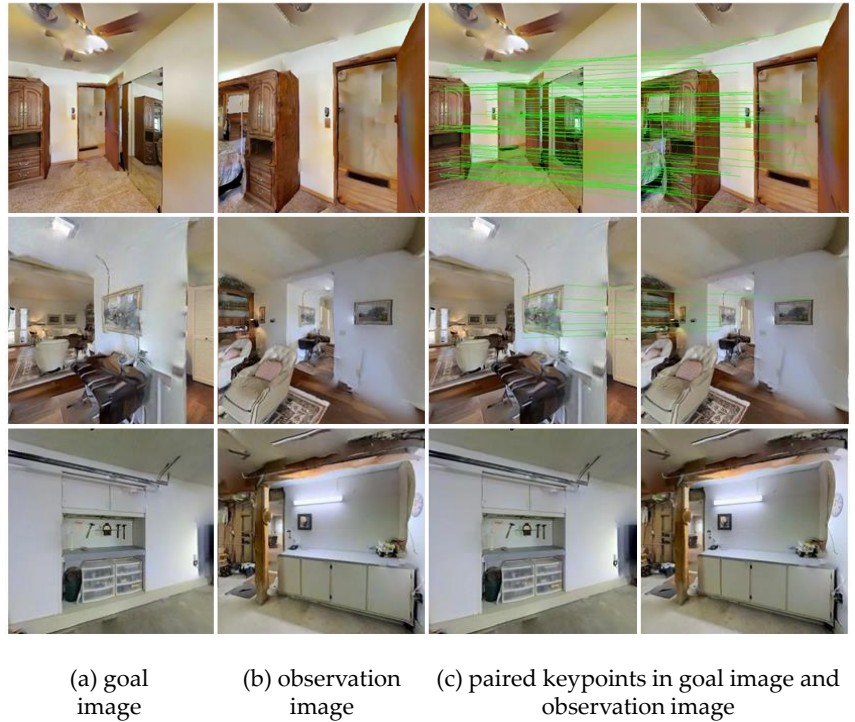

| (a) goal image | (b) observation image | (c) paired keypoints in goal image and observation image |

Figure 4: **Visualization of the matching results of FGPrompt-SF.** Paired keypoints are connected with green lines in the last two columns.

matching module does not contribute to the navigation policy, which is particularly significant in the longer episodes that start far from the goal position.

**Visualizing FGPrompt-EF.** We show the visualization result of the early fusion mechanism of our proposed FGPrompt-SF in Figure 5. The last column present EigenCAM [9] visualized activation maps from the first layer of the joint encoder backbone. Guided by the fine-grained goal prompts in the early fusion scheme, the visual backbone focuses more on regions related to the goal image.

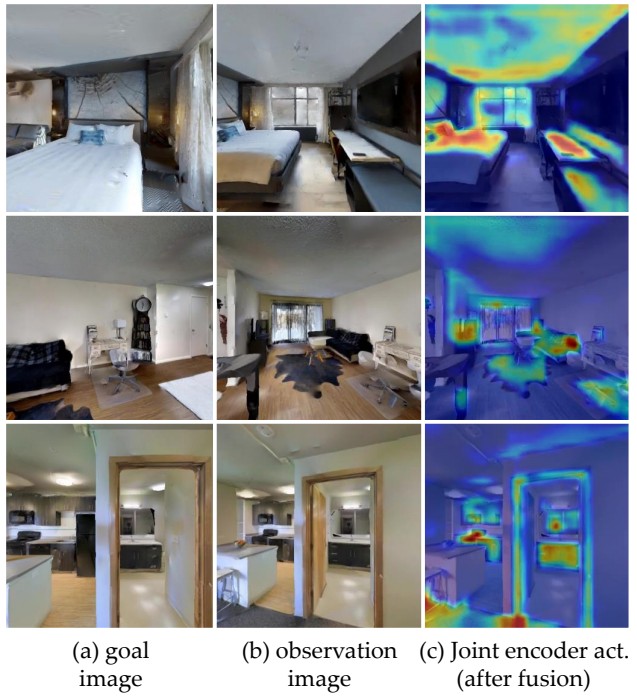

(a) goal
image

(b) observation
image

(c) Joint encoder act.
(after fusion)

Figure 5: **Visualization of the activation map of FGPrompt-EF.** We use EigenCAM [9] to reveal where the model pays more attention.

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
