# OpenReview forum: "FGPrompt: Fine-grained Goal Prompting for Image-goal Navigation"
_NeurIPS.cc/2023/Conference — NeurIPS 2023 poster_

### Official Review · Reviewer_CU6j · 2023-07-02

**Soundness:** 3 good
**Presentation:** 3 good
**Contribution:** 3 good
**Rating:** 5
**Confidence:** 4

**Summary:**

The paper proposes FGPrompt that conditions the goal image embedding on the observation such that the agent can obtain goal-relevant visual cues during image-goal navigation.
To fuse the input and goal image embeddings, FGPrompt introduces two strategies: Mid Fusion by FiLM and Early fusion by encoding the concatenation of the input and goal images.
The proposed method achieves a new state-of-the-art by large margins in the Gibson dataset.

**Strengths:**

- Mid and early fusing the input and goal images to capture goal-relevant information is well-motivated and sounds sensible.
- The proposed method achieves substantial improvements over prior arts by large margins, even with fewer parameters.
- Good illustrations of the proposed method and qualitative examples help better understand the method and its efficacy.

**Weaknesses:**

- The novelty of the proposed method is a bit weak (see Q1).
- Early Fusion learns Resnet to jointly encode the input and goal images to capture goal-relevant clues. However, it is not clear what is the difference between a simple CNN-GRU-based policy and the proposed method (see Q2).
- It is not trivial to leverage prior knowledge (e.g., pretrained large models) (Q3).

**Questions:**

- Q1: The paper motivates the necessity of fusing the input and goal images but its methodology is directly adopted from FiLM (Mid Fusion). It is unclear which part is novel in the proposed method and what we can learn from the novel part.
- Q2: The best-performing model (Early Fusion) uses Resnet to jointly encode the input and goal images, resulting in a common CNN-LSTM-based architecture. This seems not surprising as we have already been using neural networks to let them learn how to effectively encode input (in this case, the input and the goal images). Jointly encoding the input and goal images is evidenced by the authors' quantitative analyses but I'm not sure if this is novel.
- Q3: The best-performing model (Early Fusion) is trained in an end-to-end manner. However, this makes it hard for the model to leverage external knowledge, possibly from large models, as they are not usually trained with the concatenation of images. Can the proposed method leverage such knowledge?

**Limitations:**

The authors have adequately addressed the limitations.

---

> ### Author Rebuttal · Authors · 2023-08-09
>
> > Q1. The paper motivates the necessity of fusing the input and goal images but its methodology is directly adopted from FiLM (Mid Fusion). It is unclear which part is novel in the proposed method and what we can learn from the novel part.
>
>
> A1. We are sorry for the confusion. Nevertheless, **we have to clarify that our method is not directly adopted from FiLM**. We here highlight two main differences between our method and FiLM-based methods[A][B][C]:
>
> - **Different implementation details.** Existing FiLM-based methods first encoder the conditional input (a textual sentence or an image) to a semantic feature and then map this semantic feature into a single affine transformation factor with shape of 1x1xC. In the observation encoder, all its activation values in spatial dimension are affine-transformed using the same factor. In contrast, we calculate pixel-wise affine transformation factors that shape HxWxC from high-resolution feature maps, and thus the activation in different spatial positions can be transformed in a fine-grained manner.
> -  **Different targets**. Existing FiLM-based methods focus on extracting semantic information from conditional input. However, this semantic information is insufficient for the image navigation task, since the agent relies on detailed clues from the goal image (e.g., texture details and semantic categories of numerous objects) to infer the goal position relative to current observation.
>
>
> To verify the necessity of the pixel-wise affine transformation factors, we introduce a variant that uses the single affine transformation factor inferred from the semantic features of the goal image. In the table below, the FiLM approach incorporating our pixel-wise affine transformation factors showcases notably enhanced performance.
>
>
> | Affine Transormation Factor in FiLM| SR       | SPL      |
> |----------------------------|----------|----------|
> | Single [B]               | 32.0     | 24.0     |
> | Pixel-wise (Ours)               | **77.3** | **50.4** |
>
>
>
> [A] FiLM: Visual Reasoning with a General Conditioning Layer. AAAI 2018. \
> [B] BC-Z: zero-shot task generalization with robotic imitation learning. CoRL 2021. \
> [C] Using both demonstrations and language instructions to efficiently learn robotic tasks. ICLR 2023.
>
>
> > Q2. The best-performing model (Early Fusion) uses Resnet to jointly encode  the input and goal images, resulting in a common CNN-LSTM-based  architecture. This seems not surprising as we have already been using  neural networks to let them learn how to effectively encode input (in  this case, the input and the goal images). Jointly encoding the input and goal images is evidenced by the authors' quantitative analyses but  I'm not sure if this is novel.
>
>
> A2. **Our method is fundamentally different from common CNN-LSTM-based architectures [ZER, ZSON, OVRL].** These methods employ two separate CNNs to independently encode the goal image and observation image and then feed the concatenation of these encoded features into LSTM. In contrast, we pursue the integration of information from both the goal image and observation image during the encoding process by concatenating these images in pixel level and using a single CNN to encoder them.
>
> Unlike the prior models where CNNs lack access to information from the goal image while encoding observations, and vice versa, **our approach ensures a more cohesive fusion of these sources of information.** This is critical for the image navigation task because the agent relies on detailed clues from the goal image to infer the goal position relative to current observation. Experimental results in the table below also demonstrate the superiority of our method compared with the common CNN-LSTM-based architecture.
>
>
> Besides, our contributions are also recognized by the reviewer 9djz as "*the main takeaway from the paper simple but powerful. **The fact that channel concatenation for current and goal images works so well on ImageNav is something that should be known more generally.***".
>
>
> | Methods | Joint Encoder | SR       | SPL      |
> |---------|---------------|----------|----------|
> | ZER     | No            | 29.2     | 21.6     |
> | ZSON    | No            | 36.9     | 28.0     |
> | OVRL    | No            | 54.2     | 27.0     |
> | **Ours**    | **Yes**           | **92.3** | **68.5** |
>
>
>
>
>
> > Q3. The best-performing model (Early Fusion) is trained in an end-to-end  manner. However, this makes it hard for the model to leverage external  knowledge, possibly from large models, as they are not usually trained  with the concatenation of images. Can the proposed method leverage such  knowledge?
>
> A3. **Yes, our proposed method is compatible with external knowledge in pre-trained models.** To verify this, We initialize the visual encoder using pre-trained large models (i.e., CLIP-RN50) except for the first convolution layer. For the first layer, we copy the original weight and duplicate it on the input channel dimension. Then we finetune the model in an end-to-end manner for 50M steps. Experiment results show the advantage of utilizing external knowledge. We believe that the pre-trained model helps the model to learn faster and potentially provides useful experiences for finding objects in images.
>
> | Method    | External knowledge  | SR   | SPL  |
> |-----------|---------------------|------|------|
> | Ours (EF) | without knowledge | 41.1 | 26.8 |
> | Ours (EF) | with knowledge (CLIP-RN50)          | **64.3** | **29.4** |

---

> > ### Comment · Reviewer_CU6j · 2023-08-16
> > **Rebuttal acknowledgment**
> >
> > I thank the authors for addressing my concerns with a comprehensive explanation with additional experiments. The thorough response has clarified the issues I raised. I am happy to raise my rating accordingly.

---

> > > ### Author Response · Authors · 2023-08-16
> > >
> > > Thank you for your insightful comments and constructive feedback. They are invaluable to the improvement of my work.

---

### Official Review · Reviewer_mycq · 2023-07-05

**Soundness:** 3 good
**Presentation:** 3 good
**Contribution:** 3 good
**Rating:** 7
**Confidence:** 4

**Summary:**

The paper proposes different early to middle fusion mechanisms to improve performance of ImageNav tasks thanks to the availability of higher-resolution information in early to intermediate visual encoder layers. The best proposed method (also the simplest in terms of implementation) just concatenates the target and current images and jointly processes them via early fusion, correspondingly using a stem with 6 input channels. The experiments, using three random seeds, are conclusive about the performance of all proposed methods.

**Strengths:**

- Simple design (especially the early fusion proposal) enabling high-resolution spatial reasoning for ImageNav.
- Excellent performance in comparison to SOTA methods.
- In-depth analysis for the proposed methods.

**Weaknesses:**

- I wish the method had also been applied to other task types, e.g. visual rearrangement [1]. Showing good performance in a single task type is a bit limited, even if two variants (panoramic versus limited FOV) are considered.
- I understand that Mid Fusion can be seen as the most interesting in terms of ablations and visualizations, but I think it feels somewhat strange that a large portion of the discussion in the paper ends up being about the second best proposal.

[1] Weihs et al., Visual room rearrangement, CVPR 2021.

**Questions:**

- The paper states an image used as a goal is is a clearer description than language and shows a wide range of application prospects. Arguably, most of the interaction with robots is likely going to happen through natural language, which is our natural vehicle to share information, so I'd consider adding more context to the introduction.

**Limitations:**

- They're adequately addressed.

---

> ### Author Rebuttal · Authors · 2023-08-09
>
> > Q1. I wish the method had also been applied to other task types, e.g. visual  rearrangement. Showing good performance in a single task type is a  bit limited, even if two variants (panoramic versus limited FOV) are considered.
>
> A1. Thanks for your valuable suggestion. We conduct experiments on the 1-Phase track of visual rearrangement task and **find our method useful in this task**. We start from a ResNet18+IL baseline that separately encodes the unshuffled image and agent's current observation(the walkthrough image) without fusion mechanism and learn from expert actions. Then we introduce our proposed FGPrompt into the baseline model by fusing the observation with the unshuffled image in an early stage, resulting in one jointly modeled ResNet encoder. **With our FGPrompt, the agent performs much better than the baseline agent**. We believe it helps the agent to locate correspondent or inconsistent objects in the environment. We report the testing metrics on the visual rearrangement 2023 dataset.
>
> | Method  | Success↑ | FixedStrict↑ | E↓  |
> |---------|----------|--------------|-----|
> | ResNet18+IL Basline [A] |   1.89   |     4.92      | 1.32 |
> | Ours    |   **7.68**    |     **20.1**     | **0.88** |
>
> Besides, we also found that our method is useful on the instance imagenav task regarded to the experiment results in R2Q1. All these results indicate the strong ability of our proposed methods to generalize to a variety of embodied tasks.
>
> [A] Visual room rearrangement, CVPR 2021.
>
> > Q2. I understand that Mid Fusion can be seen as the most interesting in terms of ablations and visualizations, but I think it feels somewhat strange that a large portion of the discussion in the paper ends up being about the second best proposal.
>
>
> A2. Sorry for the confusion. Actually, a trade-off exists between these two methods.
>
> - The early-fusion mechanism is somehow an interesting finding in that it performs competitively and has a simpler architecture. However, though performs well on the default setting, the early-fusion does not generalize well to other scenarios that the goal camera parameters don't match with the agent's one.
> - Our delicately designed mid-fusion mechanism performs better in this case, as evidenced by the attached experimental results from the instance imagenav task in R2Q1. These results indicate that a carefully designed mid-fusion scheme with more inductive bias is necessary.
>
> We will add more discussion and clarify these findings in the revision.
>
>
>
> > Q3. The paper states an image used as a goal is a clearer description  than language and shows a wide range of application prospects. Arguably,  most of the interaction with robots is likely going to happen through  natural language, which is our natural vehicle to share information, so  I'd consider adding more context to the introduction.
>
> A3. Thanks for your comment. We will modify the statement to make it more precise.

---

> > ### Comment · Reviewer_mycq · 2023-08-14
> > **Rebuttal acknowledgment**
> >
> > I thank the authors for the time to run additional experiments (also the ones proposed by other reviewers), which in my opinion can raise the value of the already solid paper.
> >
> > Taking into account the solidity of the results and the broader scope of the paper given by the different task types, I am happy to raise my rating.

---

> > > ### Author Response · Authors · 2023-08-14
> > >
> > > Thanks again for your valuable comments!

---

### Official Review · Reviewer_9djz · 2023-07-08

**Soundness:** 4 excellent
**Presentation:** 2 fair
**Contribution:** 2 fair
**Rating:** 7
**Confidence:** 5

**Summary:**

This paper introduces FGPrompt (Fine-grained Goal Prompting) for the image-goal navigation task (ImageNav). Existing methods for ImageNav suffer from limitations in capturing detailed goal information and focusing on goal-relevant regions in observation images. FGPrompt tries out three different methods for goal prompting to overcome these limitation, including: 1) keypoint matching, 2) FiLM layers and 3) channel concatenation with 2 and 3 emerging as strong techniques for goal prompting. Experimental results on benchmark datasets demonstrate significant performance improvements compared to existing methods while using much smaller model sizes.

**Strengths:**

1) I find the main takeaway from the paper simple but powerful. The fact that channel concatenation for current and goal images works so well on ImageNav is something that should be known more generally.
2) I found the paper easy to read and follow.
3) The method transfers really well to other scene datasets compared to prior approaches.

**Weaknesses:**

In my opinion, the paper creates a more complicated story around the various fusion techniques than required. Mid fusion technique is more complicated, requires additional computation, and still performs worse than early fusion. I have two possible suggestions for the authors to make the paper more coherent:
1) Make the mid-level fusion technique a baseline and focus on understanding the early fusion technique further
2) Or, show a scenario/task where the mid-fusion technique might be more useful. I believe that the mid-fusion technique might generalize more on the instance imagenav[1] task where the goal image is assumed to be coming from a camera with different parameters to the camera on the robot.

[1] Krantz, J., Gervet, T., Yadav, K., Wang, A., Paxton, C., Mottaghi, R., ... & Chaplot, D. S. (2023). Navigating to Objects Specified by Images. arXiv preprint arXiv:2304.01192.

**Questions:**

While this is not required to be part of the rebuttal, it would be interesting to see how well the trained policy transfers to the real world given its strong transfer on the other scene datasets.

Suggestion:
1) In line 61 the authors claim that they design the early fusion mechanism by concatenating the goal and observation images. I would suggest that the claim be toned down to say that they try various ways of goal and observation concatenation.
2) The claim that OVRL-v2 uses pose information in ImageNav is incorrect. The OVRL-v2 paper states that the pose information is only used for the ObjectNav task as mentioned on Page 4.

**Limitations:**

Since the current FGPrompt technique has the assumption that the current image and goal image will be from cameras with the same parameters (resolution, FOV, etc), I will like to hear from the authors about the limitations of their work where the camera parameters don't match[1]. This is especially important given the community is moving towards these harder tasks.

---

> ### Author Rebuttal · Authors · 2023-08-09
>
> > Q1. Mid fusion technique is more complicated, requires additional computation, and still performs worse than early fusion. I believe that the mid-fusion technique might generalize more on the instance imagenav task where the goal image is assumed to be coming from a camera with different parameters to the camera on the robot.
>
>
> A1. Sorry for the confusion. **We do find that the mid-fusion technique generalizes better on harder tasks like instance imagenav**, which is difficult due to inconsistent camera parameters. We evaluate three models, namely the baseline model (seperately encode goal image and observation image), our early-fusion agent, and mid-fusion agent, on this task. All these models are trained on the Gibson ImageNav dataset and directly transfer to the HM3D instance imagenav task.
>
>
> In the table below, the baseline model performs poorly in this task with a very low success rate (less than 1%). The agents with our proposed fusion mechanisms both perform better. **We also observed that the mid-fusion variant actually outperforms early fusion in this scenario**, as its delicately designed activation deformation module yields explicit and adaptive guidance from the goal image to the observation encoder.
>
>
> | Method        | Success | SPL |
> |---------------|---------|-----|
> | Baseline (no fusion)      | 0.6     | 0.2 |
> | Ours (EF)     | 3.4     | 0.8 |
> | Ours (MF)     | **9.9** | **2.8** |
>
> We also agument the camera height, pitch and HFOV of goal image in Gibson ImageNav eval episodes to evaluate whether these models can handle the situation when goal image and observation image are captured by cameras with different parameters. Specifically, we follow the distribution of these parameters in the instance imagenav paper [A], sampling goal camera height $h\sim\mathcal{U}(0.8m,1.5m)$, pitch delta from $\mathcal{U}(-5^{\circ},5^{\circ})$, and HFOV from $\mathcal{U}(60^{\circ},120^{\circ})$.
>
>
> In the table below, **we also find that the mid-fusion mechanism performs the best in this scenario**. All these results reveal the effectiveness and robustness of mid-fusion in harder tasks.
>
> | Method        | Success  | SPL      |
> |---------------|----------|----------|
> | Baseline (no fusion)     | 12.1     | 10.5     |
> | Ours (EF)     | 64.6     | 38.5     |
> | Ours (MF)     | **78.1** | **52.7** |
>
> [A] Instance-Specific Image Goal Navigation: Training Embodied Agents to Find Object Instances. arXiv 2022.
>
> > Q2. Make the mid-level fusion technique a baseline and focus on understanding the early fusion technique further.
>
> A2. Thanks for the valuable suggestion. For the mid-fusion technique, as analyzed in Q1, it performs the best on the more complicated and practical instance navigation task. We will show its power in instance imagenav task in the revision.
>
> As for the understanding of early-fusion technique, we have conduct an analysis on it as an extension of the mid-fusion scheme in the section F in the appendix, by means of an EigenCAM visualization. We will update the manuscript to make it more clear.
>
>
> > Q3. While this is not required to be part of the rebuttal, it would be interesting to see how well the trained policy transfers to the real world given its strong transfer on the other scene datasets.
>
> A3. Thanks for your valuable comment. It is an interesting idea to apply our method on a real robot. Since setting up a real robot in a short period is difficult, we left it as our future work.
>
> > Q4. The claim that OVRL-v2 uses pose information in ImageNav is incorrect. The OVRL-v2 paper states that the pose information is only used for the ObjectNav task as mentioned on Page 4.
>
> A4. Thanks for your correction, we will modify the statement in the manuscript.
>
> > Q5. Since the current FGPrompt technique has the assumption that the current image and goal image will be from cameras with the same parameters (resolution, FOV, etc), I will like to hear from the authors about the limitations of their work where the camera parameters don't match.
>
> A5. We agree with the reviewer that the problem definition in ImageNav task has the assumption that the goal images are taken under the same camera setting with the agent and we actually train the agent in these images. As discussed in Q2, we have two main findings on our method:
>
> - As the experiment we have done in Q2, our FGPrompt outperforms baseline methods without fusion mechanism by a large margin. It reveals that our FGPrompt still shows potential in solving this harder task in contrast with baseline methods.
> - The performance of our methods on the instance imagenav task is relatively low compared to the ImageNav task. We speculate that the extremely different perspective of goal images that haven't been seen during training and a longer episode length undermine the performance of our method. This result hints us that our method could make a further improvement when combined with memory-based methods [A] and [B] to achieve more efficient large-scope exploration. We leave this as our future work.
>
> [A] Visual graph memory with unsupervised representation for visual navigation. ICCV 2021. \
> [B] Topological semantic graph memory for image-goal navigation. CoRL 2023.

---

> > ### Comment · Reviewer_9djz · 2023-08-18
> > **Thank you for the rebuttal**
> >
> > I thank the authors for the effort they put into writing the rebuttal. I am happy to see how the paper's story has evolved to be more consistent with the rebuttal. I am updating my score and I expect the authors will follow through to add a section on the strengths of the mid-fusion technique in the final manuscript.

---

### Official Review · Reviewer_o71w · 2023-07-10

**Soundness:** 3 good
**Presentation:** 2 fair
**Contribution:** 3 good
**Rating:** 7
**Confidence:** 3

**Summary:**

The authors offer a solution to the image-goal navigation task. The solution focuses on granular feature extraction from the goal image early on in the model pipeline, and usage of the goal image to inform whcih features in the observation the agent should attend to. The paper offers multiple mechanisms to do so: Skip fusion, Mid fusion and Early fusion.

**Strengths:**

* Strong results that seem to outdo the SOTA significantly
* The method illustration diagram + EigenCAM visuals were well done
* Plenty of ablations were explored.


**Weaknesses:**

* This isn't necessarily a critique of the contribution of the paper as much as it is of the narrative. The best performing method in the paper (Early Fusion), while having impressive performance, doesn't demonstrate a significantly novel method.(I believe its channel concatenation of the goal and observation before passing through a fully connected MLP. ) As mentioned in the references, image prompting has been used across many other applications as well. It may be worth writing this paper as an establishment of a new baseline of how Image Goal navigation is done, as opposed to a novel contribution.

**Questions:**

* It seems like in Fig 1 graph b, works that solve the the ObjectNav (as opposed to ImageNav) task (ZSON) are also plotted. Is there a reason for that?
* Maybe its worth specifying that FC -> fully connected MLP for eq (1)
* Was there a significant difference observed in training time/resources needed compared to the baseline for each of the methods?
* It seems to me like you are attempting to implement a form of attention. Have you thought about using self attention module but with the observation as the key and goal as the query as a form of fusion?
* It may be worth running fine-tuning experiments on OOD Datasets
* Out of curiosity, have you tried evaluating using an agent with a different height of the sensor/camera?
* I also wonder how big of a role environmental context plays, ie, what would happen if you trained with the goal images where the background was masked out? This is not a priority, just a curiosity.

---

> ### Author Rebuttal · Authors · 2023-08-09
>
> > Q1. The best performing method in the paper (Early Fusion), while having impressive performance, doesn't demonstrate a significantly novel method.
>
> A1. Thanks for your comments. We would still like to point out the novelty and contribution of our early-fusion method. We empower the ImageNav agent with the **fine-grained information exchange ability** through a simple yet powerful channel concatenation technique on the **very beginning** of the convolution network. The contributions are well recognized by the reviewer 9djz as “the main takeaway from the paper simple but powerful. The fact that channel concatenation for current and goal images works so well on ImageNav is something that should be known more generally.”.
>
> > Q2. As mentioned in the references, image prompting has been used across many other applications as well. It may be worth writing this paper as an establishment of a new baseline of how Image Goal navigation is done, as opposed to a novel contribution.
>
> A2. Existing image prompting methods [A][B] focus on extracting semantic information from conditional input. However, it is insufficient for the image navigation task, since the agent relies on detailed clues from the goal image (e.g., texture details and semantic categories of numerous objects) to infer the goal position. In contrast, **our method focuses on fine-grained promting.** We keep the spatial structure to preserve these fine-grained clues in the feature maps during the fusion schemes.
>
> To verify the necessity of fine-grained prompting, we introduce a variant that directly prompts the observation encoder using the semantic features of the goal image. In the table below, both our mid-fusion and early-fusion technique with fine-grained prompting showcases notably enhanced performance.
>
> |Setting|SR|SPL|
> |-|-|-|
> | Semantic Prompting [A][B]  | 32.0| 24.0|
> | Our Fine-grained Prompting (MF) | 77.3| 50.4|
> | Our Fine-grained Prompting (EF) | **78.9** | **54.7** |
>
> [A]  BC-Z: zero-shot task generalization with robotic imitation learning. CoRL 2021. \
> [B] Using both demonstrations and language instructions to efficiently learn robotic tasks. ICLR 2023.
>
> > Q3. It seems like in Fig 1 graph b, works that solve the ObjectNav (as  opposed to ImageNav) task (ZSON) are also plotted.
>
> A3. Sorry for the confusion. ImageNav results has also been reported in ZSON paper. To make a wide comparison among all ImageNav methods, we include it in graph b.
>
> > Q4. Maybe its worth specifying that FC -> fully connected MLP for eq (1).
>
> A4. We have modified the equation in the revised version.
>
> > Q5. Was there a significant difference observed in training time/resources needed compared to the baseline for each of the methods?
>
> A5. Yes. Our early-fusion method significantly reduces the training cost in terms of GPU hours. In comparison, the mid-fusion method slightly increase training cost as it introduces an additional FiLM layer with a convolution operator to obtain affine factors. We provide detailed numbers training a ResNet-50 agent 1x3090 GPU.
>
> | Methods| Frames per second (FPS)↑ | GPU hours↓ |
> |-|-|-|
> | Baseline | 67 | 2070|
> | Ours (MF) | 65  | 2135|
> | Ours (EF) | **126** | **1101**|
>
>
> > Q6. Have you thought about using self attention module?
>
> A6. We conduct an experiment that replaces the FiLM module with a self-attention module. Specifically, we project the flattened feature map from the first layer of goal encoder into query and the correspondent sequence from observation encoder into key and value. Then, we utilize the self-attention operation to merge the goal and observation features. Experiment results are shown below:
>
> |Methods|SR|SPL|
> |-|-|-|
> |Self-attention|13.9|12.2|
> |Ours| **77.3** | **50.4** |
>
> The results indicate the efficiency of the fine-grained conditioned reasoning through FiLM affine transformation compared to a attention mechanism that is hard to learn.
>
> > Q7. It may be worth running fine-tuning experiments on OOD Datasets
>
> A7. We finetune our FGPrompt agent on HM3D that contains different scenes structures and contents for 100M steps. **As shown in the table below, fine-tuning on HM3D further improve the SR from 76.1% to 81.9%.** Besides, our FGPrompt agent significantly outperforms the baseline by a large margin, demonstrating the effectiveness of our method.
>
> | Methods  | Pre-trained | Fine-tuned | SR   | SPL  |
> |-|-|-|-|-|
> | Baseline | Gibson| -| 9.6  | 6.3  |
> | Baseline | Gibson| HM3D| 20.2 | 17.7 |
> | Ours| Gibson| -| 76.1 | 49.6 |
> | Ours| Gibson| HM3D| **81.9** | **54.5** |
>
>
> > Q8. Have you tried evaluating using an agent with a different height of the sensor/camera?
>
> A8. Yes. We report the evaluation results under different camera heights in the following table. We found that even changing the height of the agent's camera during testing, **our method still outperforms the baseline consistently.**
>
> | Methods  | Training agent height | Eval agent height | SR   | SPL  |
> |-|-|-|-|-|
> | Baseline | 1.25 | 1.25  | 29.2 | 21.6 |
> | Baseline | 1.25 | 1.5| 12.8 | 11.3 |
> | Ours | 1.25 | 1.25| 90.7 | 62.1 |
> | Ours  | 1.25 | 1.5| 76.0 | 46.7 |
>
>
> > Q9. I also wonder how big of a role environmental context plays.
>
> A9. To investigate the importance of environmental context, we train our FGPrompt agent using background-removed goal images. We set the pixel of background (e.g., uncountable objects such as wall and floor) to zero according to the ground truth segmentation map. We leverage the semantic annotation in HM3D v2 dataset to obtain segmentation maps. Numbers are reported as below:
>
> |Setting|SR|SPL|
> |-|-|-|
> |Ours w/o background|64.4|45.2|
> |Ours w/ background|**75.9**|**48.2**|
>
> From the experimental results, our method suffered from a slight degradation when environmental context is removed from the goal image. These results indicate that environmental context in image background provides useful but limited clues. We believe that objects with their arrangement in each room play a critical role in our FGPrompt.

---

> > ### Comment · Reviewer_o71w · 2023-08-16
> > **Response to rebuttal**
> >
> > Thank you for the detailed response! No further followups from me.

---

> > > ### Author Response · Authors · 2023-08-17
> > >
> > > Thanks for your valuable comments!

---

### Author Rebuttal · Authors · 2023-08-09

We sincerely appreciate all reviewers’ time and efforts in reviewing our paper and for the constructive feedback. In addition to the response to specific reviewers, here we would like to 1) thank reviewers for their acknowledgment of our work, 2) summarize our contributions, and 3) highlight the new results added during the rebuttal:

1). We are glad that the reviewers appreciate and recognize our contributions.
1. The proposed method achieved substantial improvements. [o71w, 9djz, mycq, CU6j]
2. The main takeaway from our paper is powerful. [9djz, mycq]
3. The proposed method transfer really well. [9djz]
4. The proposed method is well-motivated. [CU6j]
5. The ablation studies are in-depth with good visualization. [o71w, mycq, CU6j]
6. This paper is well-written and easy to follow. [9djz]

2). We summarize our contributions as follows.
- **A novel image goal prompting architecture to solve the ImageNav task.** We propose a fine-grained image goal prompting (FGPrompt) architecture to explicitly exchange fine-grained information between goal image and observation image, reaching a new SOTA of the ImageNav task and also showing great potential in some other embodied tasks including instance image navigation and visual rearrangement tasks.
- **A simple early-fusion scheme to boost the ImageNav performance with very few parameters.** We empower the ImageNav agent with the fine-grained information exchange ability through a simple yet powerful channel concatenation technique. This scheme shows an absolute advantage on the ImageNav task even compared to some complex memory graph-based methods.
- **A generic mid-fusion scheme to address the mismatch between goal camera and observation camera.** We dedicately design a mid-fusion scheme through a novel fine-grained FiLM mapping module to perform more robust information exchange. This scheme shows superior performance in more practical scenarios that the goal image possesses different camera parameters from the observation.
- **An in-depth analysis of the image prompting schemes.** We illustrate the activation maps using EigenCAM and reveal how mid-fusion and early-fusion scheme work, bringing new insights for the embodied AI field.

3). In this rebuttal, we have added more supporting results following the reviewers’ suggestions.
1. Comparison results with an attention-based mid-fusion model. [o71w]
2. Finetuning results on the OOD dataset. [o71w]
3. Training efficiency by means of GPU hours compared with baseline. [o71w]
4. Evaluation under different camera settings. [o71w]
5. Ablation results of background context. [o71w]
6. Evaluation on instance image navigation task. [9djz]
7. Evaluation on visual rearrangement task. [mycq]
8. Performance of our method with pre-trained CLIP initialization. [CU6j]

---

### Comment · Area_Chair_4N3h · 2023-08-11
**Discussion with authors**

Dear Reviewers,

Please check the rebuttal and start a discussion with authors if you need any additional information to make your final decision. The discussions should be completed by Aug 16.

Thanks,

AC

---

### Author Response · Authors · 2023-08-13

Dear AC and all reviewers:

Thanks again for all the insightful comments and advice, which helped us improve the paper's quality and clarity.

The discussion phase has been on for several days and we have not heard any post-rebuttal responses yet.

We would love to convince you of the merits of the paper. Please do not hesitate to let us know if there are any additional experiments or clarification that we can offer to make the paper better. We appreciate your comments and advice.

Best,
Author

---

### Decision · Program_Chairs · 2023-09-21

**Decision:**

Accept (poster)

**Comment:**

The paper proposes an approach for the image-goal navigation task. The reviewers have strongly supported the paper since: (1) The method achieves strong results compared to the state-of-the-art approaches. (2) The method is simple and intuitive. (3) The method transfers well to other scene datasets.
(4) The paper is easy to follow and motivates the approach very well. (5) The paper includes in-depth analysis of the models.

Therefore, the AC follows the recommendation of the reviewers and recommends acceptance.